# Learning to Estimate Single-View Volumetric Flow Motions without 3D Supervision

**Aleksandra Franz**
Technical University of Munich (TUM)
`franzer@in.tum.de`

**Barbara Solenthaler**
ETH Zurich
TUM - Institute for Advanced Study
`solenthaler@inf.ethz.ch`

**Nils Thuerey**
Technical University of Munich (TUM)
`nils.thuerey@tum.de`

## Abstract

We address the challenging problem of jointly inferring the 3D flow and volumetric densities moving in a fluid from a monocular input video with a deep neural network. Despite the complexity of this task, we show that it is possible to train the corresponding networks without requiring any 3D ground truth for training. In the absence of ground truth data we can train our model with observations from real-world capture setups instead of relying on synthetic reconstructions. We make this unsupervised training approach possible by first generating an initial prototype volume which is then moved and transported over time without the need for volumetric supervision. Our approach relies purely on image-based losses, an adversarial discriminator network, and regularization. Our method can estimate long-term sequences in a stable manner, while achieving closely matching targets for inputs such as rising smoke plumes.

## 1 Introduction

Estimating motion is a fundamental problem, and is studied for a variety of settings in two and three dimensions (Ranjan & Black, 2017; Hur & Roth, 2021; Gregson et al., 2014). It is also a highly challenging problem, since the motion $\mathbf{u}$ is a *secondary* quantity that typically can't be measured directly and has to be recovered from changes observed in transported markers $\rho$. We focus on volumetric, momentum-driven materials like fluids, where in contrast to the single-step estimation in optical flow (OF), motion estimation typically considers multiple coupled steps to achieve a stable *global transport*. Furthermore, in this setting the volume distribution of markers $\rho$ is usually unknown and needs to be reconstructed from the observations in parallel to the motion estimation.

So far, most research is focused on the reconstruction of single scenes. Classic methods use an optimization process working with an explicit volumetric representation (Eckert et al., 2019; Zang et al., 2020; Franz et al., 2021) while some more recent approaches optimize single scenes with neural fields (Mildenhall et al., 2020; Chu et al., 2022). As such an optimization is typically extremely costly, and has to be redone for each new scene, training a neural network to infer an estimate of the motion in a single pass is very appealing. Similar to most direct optimizations, existing neural network methods rely on multiple input views to simplify the reconstruction (Qiu et al., 2021). However, this severely limits the settings in which inputs can be captured, as a fully calibrated lab environment is often the only place where such input sequences can be recorded.

The flexibility of motion estimation from single views makes them a highly attractive direction, and physical priors in the form of governing equations make this possible in the context of fluids (Eckert et al., 2018; Franz et al., 2021). Nonetheless, despite using strong priors, the single viewpoint makes it challenging to adequately handle the otherwise fully unconstrained depth dimension. We target a deep learning-based approach where a neural network learns to represent the underlying motion structures, such that almost instantaneous, single-pass motion inference is made possible without relying on ground truth motion data. The latter is especially important for complex volumetric

motions, as reference motions of real fluids can not be acquired directly. Instead, one has to work with reconstructions or even simulated data, suffering from a mismatch between the observations and the synthetic motion data. While obtaining multiple calibrated captures for training is feasible, using additional views only for losses results in issues with the depth ambiguity of single-view inputs.

In this work, we address the challenging problem of training neural networks to infer 3D motions from monocular videos in scenarios where no 3D reference data is available. To the best of our knowledge, we are the first to propose an end-to-end approach, denoted by *Neural Global Transport* (NGT) in the following, which

    (i) yields a neural network to estimate a global, dense 3D velocity from a single-view image sequence without requiring any 3D ground truth as targets. Among others, this is made possible by a custom 2D-to-3D UNet architecture.

    (ii) We address the resulting depth-ambiguity problem using a new approach with differentiable rendering and an adversarial technique.

    (iii) A single network trained with the proposed approach generalizes across a range of different inputs, vastly outperforming optimization-based approaches in terms of performance.

## 2   RELATED WORK

**Optical flow**   Flow estimation is of great interest in a multitude of settings, from 2D optical flow over scene flow to the capture and physically accurate reconstruction of volumetric fluid flows. Operating on a pair of 2D images, optical flow estimates a motion that maps one to the other (Sun et al., 2014). In this setting, multi-scale approaches in the form of spatial pyramids are a longstanding technique to handle displacements of different scales (Glazer, 1987). More recent CNN-based methods also employ spatial pyramids (Dosovitskiy et al., 2015; Ranjan & Black, 2017) and can learn to estimate optical flow in an unsupervised fashion (Ahmadi & Patras, 2016; Yu et al., 2016; Luo et al., 2021).

**Scene flow**   Scene flow (Vedula et al., 1999) bridges the gap to 3D where earlier approaches using energy minimization or variational methods(Zhang & Kambhamettu, 2001; Huguet & Devernay, 2007) have been surpassed by CNNs that bring superior runtime performance while retaining state-of-the-art accuracy (Ilg et al., 2018; Saxena et al., 2019) and can also be trained without the need for ground truth data (Lee et al., 2019; Wang et al., 2019). Flow estimation from a single input is of particular importance as it vastly simplifies the data acquisition and several methods for monocular scene flow have been proposed (Brickwedde et al., 2019; Yang & Ramanan, 2020; Luo et al., 2020). These can be un- or self-supervised (Ranjan et al., 2019; Hur & Roth, 2020) and benefit from using multiple frames (Hur & Roth, 2021).

**Fluid flow**   Fluid flows are traditionally extremely difficult to capture and methods ranging from Schlieren imaging Dalziel et al. (2000); Atcheson et al. (2008; 2009) and particle imaging velocimetry (PIV) methods Grant (1997); Elsinga et al. (2006); Xiong et al. (2017) over laser scanners Hawkins et al. (2005); Fuchs et al. (2007) to structured light Gu et al. (2013) and light path Ji et al. (2013) approaches all require specialized setups. The use of multiple commodity cameras simplifies the acquisition (Gregson et al., 2014; Eckert et al., 2019) and allows for view-interpolation to create additional constrains (Zang et al., 2020). Few works have attempted to solve monocular flow estimation in the fluids setting. For single-scene optimization Eckert et al. (2018) constrain the motion along the view depth, while Franz et al. (2021) use an adversarial approach to regularize the reconstruction from unseen views. Qiu et al. (2021) have proposed a network that can estimate a long-term motion from a single view, but still require 3D ground truth for training.

**3D reconstruction**   In the context of fluids it is common to also reconstruct an explicit representation of the transported quantities. Such 3D reconstruction is typically addressed for clearly visible surfaces (Musialski et al., 2013; Koutsoudis et al., 2014) where some of the algorithms that have been proposed can incorporate deformations (Zang et al., 2018; Kato et al., 2018). In this context, volumetric reconstructions make use of voxel grids (Papon et al., 2013; Moon et al., 2018), or more recently neural network representations Sitzmann et al. (2019a); Lombardi et al. (2019); Sitzmann

et al. (2019b); Mildenhall et al. (2020). Learned approaches have likewise been used to recover volumetric scene motions, e.g., for moving human characters (Mescheder et al., 2019) or to separate static and dynamic parts (Chu et al., 2022). For coupling with visual observations, several differentiable rendering frameworks have been proposed (Kato et al., 2020; Zhang et al., 2021). These where used to constrain unseen views of static volumes (Henzler et al., 2018; 2019) and to reconstruct objects from single images as SDF (Jiang et al., 2020). Considering fluids, corresponding approaches where used in the context of SPH Schenck & Fox (2018), for the initialization of a water wave simulator Hu et al. (2020), to realize style transfer onto 3D density field Kim et al. (2019a; 2020), and for monocular flow reconstruction (Franz et al., 2021).

## 3 METHOD

In this work we focus on the task of estimating a global 3D motion $\mathbf{u}$ of a volumetric quantity $\rho$ over time from a sequence of single view images $\hat{I}^t$. To that end, we train a generator network $\mathcal{G}_{\mathbf{u}}$ to estimate the motion $\mathbf{u}^t$ between two consecutive time steps $\rho^t$ and $\rho^{t+1}$. In this context, a simple, supervised method would match the output to a reference velocity $\hat{\mathbf{u}}^{t+1}$ given a cost function such as an $\mathcal{L}_2$ loss:

$$\min_\theta \left|\left| \mathcal{G}_{\mathbf{u}}(\rho^t, \rho^{t+1}; \theta) - \hat{\mathbf{u}}^{t+1} \right|\right|^2 . \tag{1}$$

However, as ground truth velocity targets are often not available, unsupervised methods instead leverage a model for the underlying transport process. Given a differentiable discretization of the transport $\mathcal{A}$ we can learn the motion in an unsupervised fashion by requiring that $\rho^t$ should match $\rho^{t+1}$ when $\mathcal{A}$ is applied:

$$\min_\theta \left|\left| \mathcal{A}(\rho^t, \mathcal{G}_{\mathbf{u}}(\rho^t, \rho^{t+1}; \theta)) - \rho^{t+1} \right|\right|^2 . \tag{2}$$

As the image sequence $\hat{I}$ represents projections of volumes that leave the formulation of equation 2 to be highly under-constrained, we impose further priors to ensure the training converges to desirable, unimodal solutions. Assuming that an initial state $\rho^0$ is known, we define a multi-step transport as $\rho^t = \mathcal{A}^t(\rho^0, \mathbf{u}) = \mathcal{A}(\dots \mathcal{A}(\rho^0, \mathbf{u}^0) \dots, \mathbf{u}^{t-1})$, and the target $\rho^{t+1}$ above is replaced by an image-based loss from unsupervised 3D reconstruction (Zang et al., 2020; Franz et al., 2021). This loss is realized via a differentiable image formation model $\mathcal{R}$ such that the inferred volumetric state can be compared to the observed target image $\hat{I}^{t+1}$. In line with $\mathcal{R}$, the network $\mathcal{G}_{\mathbf{u}}$ is likewise conditioned on the target image $\hat{I}^{t+1}$, which yields the combined learning objective for our approach:

$$\min_\theta \left|\left| \mathcal{R}(\mathcal{A}(\rho^t, \mathcal{G}_{\mathbf{u}}(\rho^t, \hat{I}^{t+1}; \theta))) - \hat{I}^{t+1} \right|\right|^2 . \tag{3}$$

While this formulation has the central advantage that it yields a globally coupled, end-to-end process that does not require any volumetric targets, its under-constrained nature leads to a very challenging environment for learning tasks: In addition to the well-studied aperture problem of optical flow Beauchemin & Barron (1995), the 3D nature of our setting introduces the problem of *depth-ambiguity*: A single density observed in a pixel of $\hat{I}^{t+1}$ can be satisfied by an arbitrary distribution of densities along the corresponding viewing ray direction. In contrast to the aperture problem, simple regularizers such as smoothness do not suffice to address depth ambiguity. Below, we explain our solution which uses an adversarial architecture.

### 3.1 DEPTH AMBIGUITY

For each target color along a ray, an infinite number of distributions of density along this ray exists, e.g. larger and further away vs. smaller and closer. The existence of multiple parallel observations, such as the pixels of an image, likewise does not guarantee uniqueness. One approach to address this issue is to refer to multiple observations in the loss during training, as would be available from a calibrated multi-view capture setup. However, since the network has no way to infer the unique density distributions pertaining to these side targets from its single input view, such an approach typically leads to learning an suboptimally averaged solution. Instead, we address this depth-ambiguity by expanding the simple target loss

$$\mathcal{L}_{\hat{I}} = \left|\left| \mathcal{R}(\rho^{t+1}) - \hat{I}^{t+1} \right|\right|^2 \tag{4}$$

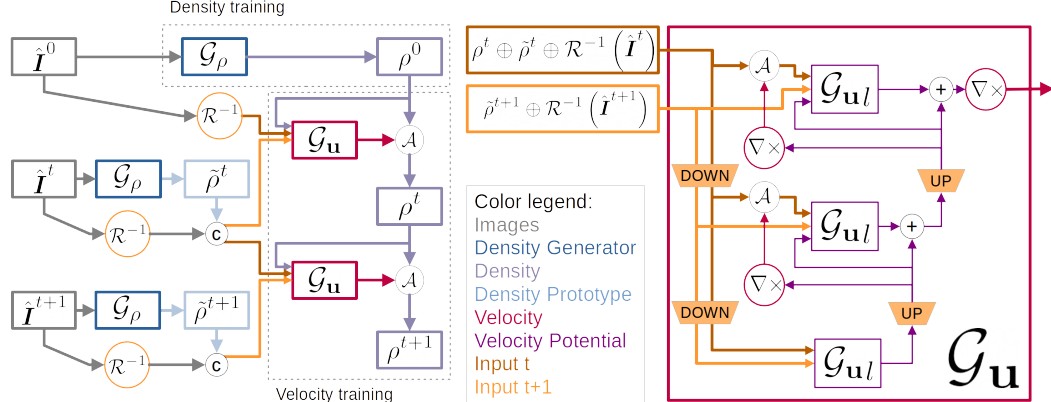

Figure 1: Left: An overview over the complete NGT framework. We generate an initial density volume $\rho^0$ that is advected by the velocity to form a sequence. Density estimates are used in addition to the single input image to guide and stabilize the velocity generation. Velocity training is done end-to-end over the whole sequence. Right: Our multi-scale velocity estimator $\mathcal{G}_{\mathbf{u}}$, shown for 3 resolution scales. The inputs contain information about the current $(t)$ and next $(t+1)$ time step. Each scale generates a residual velocity potential which is used to advect the inputs of step $t$ before generating the next residual. The final velocity is divergence free due to using the curl $\nabla\times$.

from equation 3, where $\rho^{t+1} = \mathcal{A}(\rho^t, \mathcal{G}_{\mathbf{u}}(\rho^t, \hat{\boldsymbol{I}}^{t+1}; \theta))$, with the adversarial loss used by Henzler et al. (2019) and Franz et al. (2021) to restrict $\rho$ to a plausible appearance without imposing a specific solution. Specifically, we follow previous work (Franz et al., 2021) and make use of a discriminator $\mathcal{D}$ with an RaLSGAN objective (Jolicoeur-Martineau, 2019)

$$\mathcal{L}_{\mathcal{D}}(\rho^{t+1}, l) = \mathbb{E}_{p_{\text{data}}}\left[\left(\mathcal{D}(\hat{\boldsymbol{I}}^{t+1}) - \mathbb{E}_{p_{\text{view}}(\omega)}\mathcal{D}(\mathcal{R}(\rho^{t+1}, \omega)) - l\right)^2\right]$$
$$+ \mathbb{E}_{p_{\text{view}}(\omega)}\left[\left(\mathcal{D}(\mathcal{R}(\rho^{t+1}, \omega)) - \mathbb{E}_{p_{\text{data}}}\mathcal{D}(\hat{\boldsymbol{I}}^{t+1}) + l\right)^2\right], \tag{5}$$

where $\omega$ are randomly sampled view directions and the label $l$ is 1 or -1 when training $\mathcal{D}$ or using it as loss, respectively. To focus on compact volumes and to prevent unnecessary scattering of densities along the viewing direction, we add a depth-Loss $\mathcal{L}_z$, assuming that the observed densities are concentrated around position $c_z$ along the view's depth direction, the z-axis in our case. Thus, for every position $p$ we regularize $\rho$ depending on the distance to a center position $c_z$

$$\mathcal{L}_z = \rho^{t+1}(p)^2((c_z - p_z)2/r)^2, \tag{6}$$

where $p_z$ is the projection of $p$ on the primary view direction. As we choose $c_z$ to be in the center of the grid, we normalize with half the grid resolution $r$. To further constrain and stabilize the estimated velocity we introduce a prototype volume $\tilde{\rho}^{t+1}$ generated from a single input image (see section 3.3) as 3D target for the transported density. This again takes the form of a simple $\mathcal{L}_2$ loss:

$$\mathcal{L}_{\tilde{\rho}} = \left|\left|\rho^{t+1} - \tilde{\rho}^{t+1}\right|\right|^2. \tag{7}$$

### 3.2 Estimating Volumetric Motions

In addition to the current state $\rho^t$ and the target image $\hat{\boldsymbol{I}}^{t+1}$, as indicated by equation 3, our generator network also receives $\hat{\boldsymbol{I}}^t$, $\tilde{\rho}^t$ and $\tilde{\rho}^{t+1}$ as input. The images are projected into the volume using inverse ray-marching denoted by $\mathcal{R}^{-1}$, before being concatenated to the density fields. This way, the inputs trivially match the target and rendering losses, are coherent between frames, and the network can compare the back-projections and prototype volumes to estimate the 3D velocity based on the current state $\rho^t$. Thus, the velocity estimation becomes

$$\mathbf{u}^t = \mathcal{G}_{\mathbf{u}}(\rho^t, \mathcal{R}^{-1}(\hat{\boldsymbol{I}}^t), \tilde{\rho}^t, \mathcal{R}^{-1}(\hat{\boldsymbol{I}}^{t+1}), \tilde{\rho}^{t+1}). \tag{8}$$

The architecture of $\mathcal{G}_{\mathbf{u}}$ further follows established multi-scale approaches of optical flow and scene flow (Ranjan & Black, 2017), as shown in figure 1. A residual velocity $\mathbf{u}^l$ is generated at several resolution scales $l$ with the per-resolution network component denoted as $\mathcal{G}_{\mathbf{u}l}$. Our $\mathcal{G}_{\mathbf{u}}$ further utilizes

weight sharing between scales (Luo et al., 2021), such that self-similar structures can be reused. In combination, velocity generation for a single time step takes the recursive form of

$$\mathbf{u}_l^t = \mathcal{G}_{\mathbf{u}l}(\mathcal{A}(_\downarrow\rho^t \oplus \mathcal{R}^{-1}(\boldsymbol{I}^t) \oplus_\downarrow \tilde{\rho}^t, ^\uparrow\mathbf{u}_{l-1}^t), \mathcal{R}^{-1}(\boldsymbol{I}^{t+1}), \tilde{\rho}^{t+1}, ^\uparrow\mathbf{u}_{l-1}^t) + ^\uparrow\mathbf{u}_{l-1}^t, \qquad (9)$$

where $\oplus$ is concatenation along the channel dimension, $^\uparrow\mathbf{u}_{l-1}^t$ is the up-sampled velocity of the previous scale, $\mathbf{u}_0^t = 0$ and final velocity $\mathbf{u}_L^t = \mathcal{G}_{\mathbf{u}L}(\dots)$, $^\uparrow$ denotes up-sampling using quadratic B-spline kernels, and $_\downarrow$ is down-sampling using average pooling. The inputs from the current time-step $t$ are warped such that $\mathcal{G}_{\mathbf{u}l}$ only needs to estimate the residual velocity.

Even with the constraints of equations 4 to 7, considerably large solution spaces exist that reach the target state, e.g., unphysical solutions that create mass. For this reason we include additional constraints on magnitude, smoothness and divergence. For stability we enforce the CFL condition on the residual of each velocity scale with an $\mathcal{L}_2$ loss on velocity magnitude, while smoothness is constrained for the combined velocity on each scale. To respect mass conservation the velocity field has to be divergence free, i.e. $\nabla \cdot \mathbf{u} = 0$. Instead of adding a divergence-freeness loss like most previous work, we use a hard constraint (Kim et al., 2019b) that is compatible with the residual multi-scale approach: we define $\mathbf{u}$ as the curl $(\nabla\times)$ of a vector potential as $\mathbf{u} = \nabla \times \mathcal{G}_{\mathbf{u}}$. As the curl is a linear operator, $\nabla \times (sF + G) = s\nabla \times F + \nabla \times G$, with $F, G$ being vector fields and $s$ a scalar, Equation 9 and the output of $\mathcal{G}_{\mathbf{u}}$ are treated as a vector potential.

### 3.3 DENSITY ESTIMATION

Our motion estimation additionally hinges on a second network $\mathcal{G}_\rho$, whose role it is to estimate individual volumetric densities $\tilde{\rho}^t$ from a single image $\hat{\boldsymbol{I}}^t$:

$$\tilde{\rho}^t = \mathcal{G}_\rho(\hat{\boldsymbol{I}}^t; \Phi). \qquad (10)$$

We employ this network for the prototype density loss 7 and to infer the initial distribution of markers with $\rho^0 := \tilde{\rho}^0 = \mathcal{G}_\rho(\hat{\boldsymbol{I}}^0; \Phi)$, which was previously assumed to be known. As the challenges of the single view estimation are similar for density and for velocity, using a generator network in the already established GAN setting is a natural choice and we can reuse the losses as described above: $\mathcal{L}_{\mathcal{G}_\rho} = \mathcal{L}_{\hat{\boldsymbol{I}}} + \mathcal{L}_{\mathcal{D}}(\mathcal{G}_\rho, -1) + \mathcal{L}_z$. $\mathcal{G}_\rho$ employs a custom mixed 2D/3D UNet-architecture. The encoder processes 2D information, while the skip connections are realized via $\mathcal{R}^{-1}$.

### 3.4 IMPLEMENTATION AND TRAINING

We implement NGT with TensorFlow in Python. The $\mathcal{R}$ and $\mathcal{A}$ operations are custom: For $\mathcal{R}$, used in both the target loss (equation 4) and $\mathcal{L}_{\mathcal{D}}$, we use a differentiable volume ray-marching (Franz et al., 2021), which supports attenuation, single-scattering, and the use of background images, but is limited to isometrically scattering densities and fixed lighting. For the advection operator $\mathcal{A}$ we employ a discrete, differentiable MacCormack advection scheme. First, we train $\mathcal{G}_\rho$ alone on individual frames, where we grow the resolution from 8 to 64 during training. Then we freeze $\mathcal{G}_\rho$ and train $\mathcal{G}_{\mathbf{u}}$ with up to 5 advection-steps. We again grow the resolution of the velocity estimation network, while the estimation of $\rho^0$ and the advection happen on the maximum resolution, using up-scaled velocities until $\mathcal{G}_{\mathbf{u}}$ reaches the maximum resolution. $\mathcal{D}$ is trained during both density and velocity training. We refer to appendix A for more details.

## 4 EVALUATION

We first present two ablations to illustrate the importance of handling depth ambiguity, and of the prototype density volumes. We then evaluate the method in comparison to a series of learned and optimization-based methods for a synthetic and a real-world dataset.

### 4.1 DEPTH AMBIGUITY

In Figure 2 we show the effect of depth ambiguity. If there is no depth variation, a network can easily recover the object when using multiple target views in the loss (column 2), leading to low volumetric and perceptual errors (table 1). If the object's position is randomized (column 3),

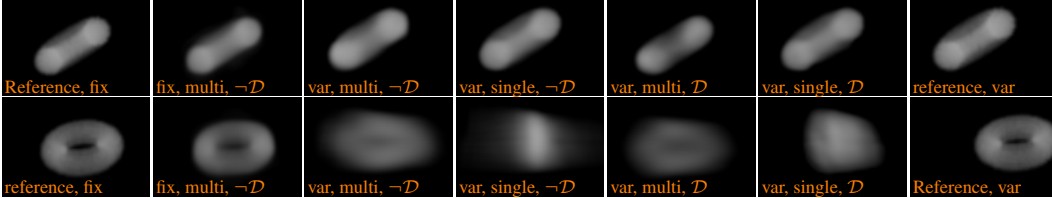

Figure 2: Depth-ambiguity ablation study with the shape dataset: In the top row all versions closely match the input view while the side view (bottom) shows severe degradation when increasing depth ambiguity by varying the position of the objects (fix → var) or using only the input view for $\mathcal{L}_{\hat{I}}$ (multi → single). Providing additional views alongside the discriminator does not yield further improvements. $\mathcal{D}$ by itself can recover a plausible configuration, even if the density does not exactly match the unknown reference location.

the network can no longer settle on a clear position as the added variation in depth is not evident from the single input. Thus, the network is presented with multiple solutions for the same input over the course of the training, severely deteriorating the quality and drastically increasing the FID by 2.5 times. Simply removing the multiple targets improves the FID at the cost of

Table 1: Quantitative evaluation of the depth-ambiguity issue as seen in figure 2. Side uses 7 evenly distributed views. Lower values are better for all metrics under consideration.

| Version | Input RMSE | Side RMSE | LPIPS | FID | $\rho$ RMSE |
|---|---|---|---|---|---|
| fix, multi, $\neg\mathcal{D}$ | .0080 | .0106 | .0200 | 48 | .260 |
| var, multi, $\neg\mathcal{D}$ | .0160 | **.0509** | .1237 | 120 | **.636** |
| var, single, $\neg\mathcal{D}$ | **.0042** | .0681 | .1345 | 98 | .969 |
| var, multi, $\mathcal{D}$ | .0151 | .0525 | .1239 | 80 | **.642** |
| var, single, $\mathcal{D}$ | .0064 | .0694 | **.1140** | **68** | .930 |

LPIPS, volume error, and visual quality. Adding the discriminator to the multi-target setup improves the perceptual metrics, but visually the result still looks blurry. Keeping $\mathcal{D}$ but removing the multiple targets further improves the perceptual quality, sharpening the results and improving FID to 68, at the cost of volumetric accuracy. In the end, the multiple target views hinder the discriminator and lead to averaged solutions that are favoured by simple metrics such as RMSE, but have lower perceptual scores. While the discriminator can not recover the true shape and depth location of the sample, as indicated by the relatively high volume error, the compact density it produces nonetheless represents a very good starting point for the motion estimation, given the single view input.

## 4.2 PROTOTYPE VOLUMES $\tilde{\rho}$

Using the prototype densities $\tilde{\rho}$ as explicit 3D constraint in the form of $\mathcal{L}_{\tilde{\rho}}$ may at first seem counter-intuitive in the presence of the depth ambiguity. However, as the $\tilde{\rho}$ density volumes are generated from single images, there is only a single $\tilde{\rho}$ associated with each input. This yields a reduction of ambiguity for the task of $\mathcal{G}_{\mathbf{u}}$ when using the prototypes as loss. Furthermore, we provide $\tilde{\rho}^{t+1}$ as additional input to $\mathcal{G}_{\mathbf{u}}$, to simplify the task of motion inference. Overall, this highlights the importance and central role of the novel 2D-to-3D UNet architecture. The addition of $\tilde{\rho}$ to the velocity training stabilizes the inference of long sequences, as evident from figure 3 and table 3. While there is not much difference in the early frames, the network lacking the stabilization clearly diverges after evaluating 120 frames.

## 4.3 RESULTS

We evaluate our method on both synthetic smoke flows and the real-world captures from the ScalarFlow dataset (Eckert et al., 2019). As a representative of simpler network architectures, i.e. without multi-scale handling and the 2D-to-3D UNet, we combine the architecture from Qiu et al. (2021) with our loss formulation. This combination is denoted by (*RapidGen*). We further compare to the direct optimization algorithms of Eckert et al. (2019) (*ScalarFlow*) and Franz et al. (2021) (*GlobTrans*). These can achieve better target matching and long-term transport because each scene is optimized over the full trajectory individually, but this comes at the cost of vastly increased (more than ×100) reconstruction times. Due to the large space of solutions for the single view problem in

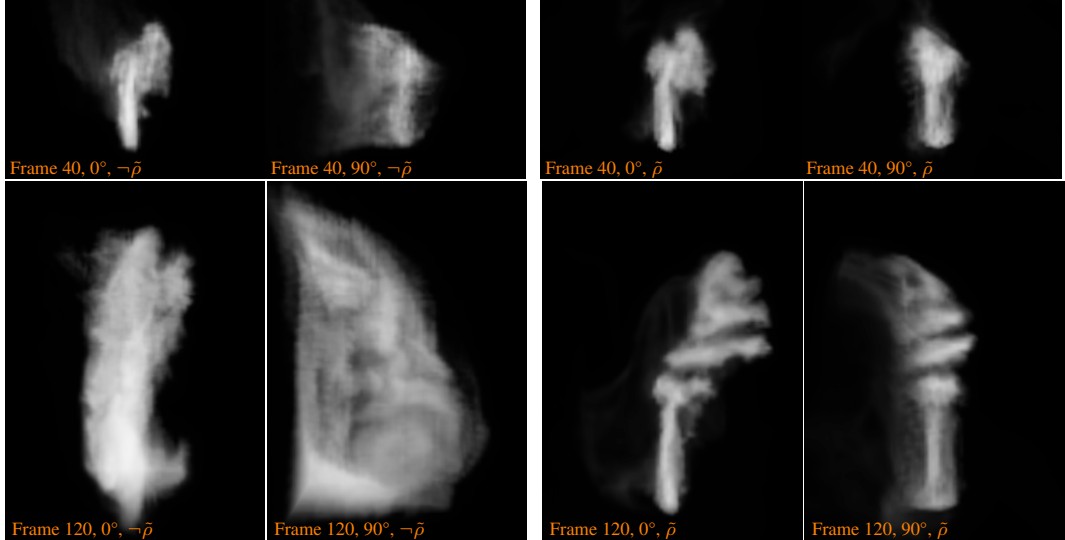

Figure 3: Adding the prototype volume $\tilde{\rho}$ produced by our 2D-to-3D UNet as guidance (right side) stabilizes the inference over long evaluations and results in better target matching and a better overall density distribution, especially from unseen views. Metrics can be found in table 3.

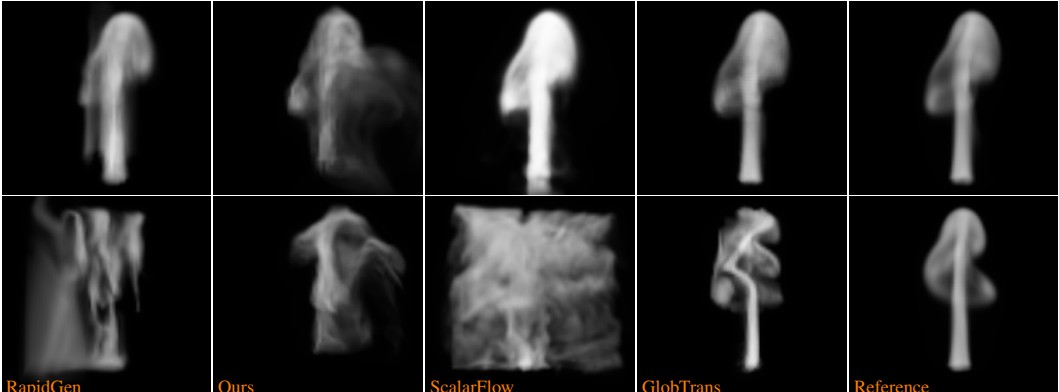

Figure 4: Qualitative comparison between different approaches using synthetic plume data for time-step 80. Top is the input view which all method match fairly well. Bottom is a 90° side view where the shortcomings of the different approaches become visible. Due to the overshoot, ScalarFlow densities are shown with a factor or $1/2$.

a setting as chaotic as fluid dynamics, we evaluate the general appearance of the estimations with the perceptual metric LPIPS (Zhang et al., 2018) and the Fréchet Inception Distance (FID) between random views of the estimated sequences and random samples of input images to obtain a measure of likeness in appearance. This allows us to asses how realistic the overall look of the plumes is.

### 4.3.1 SYNTHETIC PLUMES

We first evaluate our method with a synthetic data set consisting of simulated plumes of hot marker densities in front of a black background. While the flows are relatively simple compared to their real-world counterparts, the smooth density content makes it very difficult to estimate motions as the differentiable advection relies on spatial gradients in the density for back-propagation. For completeness, we measure the accuracy of reconstructing the input image in column one of table 2, for which all methods show a low error. Our goal is to estimate plausible solutions, rather than an exact match to a reference, as the ambiguity in depth allows for many solutions that match the available inputs. This can be seen particularly well in the volumetric comparisons, where our result has a

Table 2: Quantitative evaluation of different methods trained on a synthetic plumes dataset.

| Algorithm | Input RMSE | Side RMSE | LPIPS | FID | Random FID | $\rho$ RMSE | $\mathbf{u}$ RMSE | EPE |
|---|---|---|---|---|---|---|---|---|
| RapidGen | .0145 | .0277 | .0450 | 105 | 76 | 1.65 | .494 | .851 |
| Ours | .0156 | .0243 | .0444 | 101 | 97 | .431 | 2.06 | 3.29 |
| ScalarFlow | .116 | .120 | .158 | 135 | 106 | 1.39 | .120 | .104 |
| GlobTrans | .0122 | .0440 | .0877 | 87 | 30 | .373 | .143 | .137 |

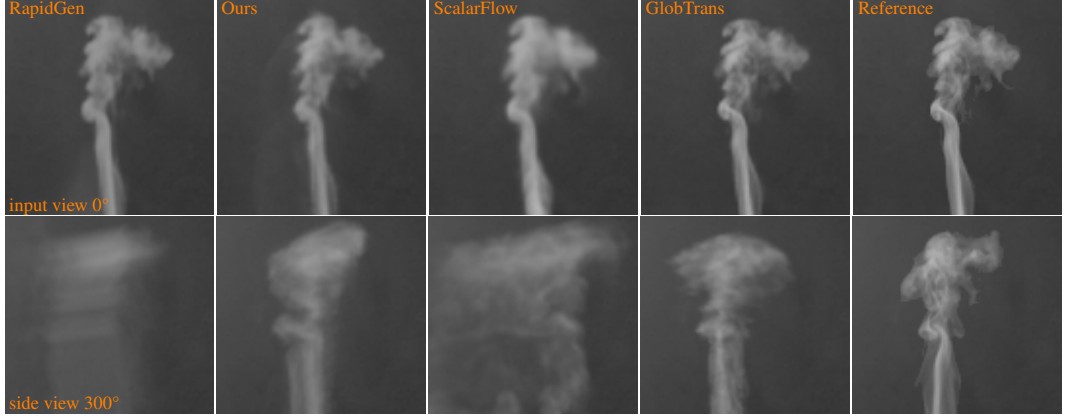

Figure 5: Qualitative comparison between different approaches using ScalarFlow data for time-step 100. Our method closely matches the given input and has a clearly defined shape that matches the general shape of the reference. It is only surpassed by the costly single-scene reconstruction method GlobTrans. RapidGen was adapted to be trained without 3D GT.

good match with the reference density with a RMSE of 0.431, but disagrees in terms of underlying motion, leading to a high difference in the velocities. The corresponding videos still show a calm, swirling motion similar to the reference. A visual comparison is shown in figure 4.

Comparing the unseen side targets reveals the differences between the approaches, especially in conjunction with the estimated motions, which can be seen in the supplemental videos. GlobTrans performs best in terms of the perceptual metrics, reaching a low FID of 30 due to having the same large smooth structures as the reference. However, GlobTrans exhibits unnatural, excessive motion in the side view. RapidGen is closer in appearance to the reference, but lacks a clearly defined plume shape leading to a density error 3.8 times higher than ours. And even though the velocity of RapidGen matches the reference better, the videos show that it consists mainly of a static upwards motion. ScalarFlow exhibits smooth and natural motions but suffers from missing depth regularization, resulting in an excessive amount of density. These results highlight the complexity of the single-view motion estimation and the multitude of viable solutions when given only a single image even for synthetic inputs, and illustrate the behavior of the different learned and optimization-based methods. Next, we evaluate their performance with real-world inputs.

Table 3: Quantitative evaluation of different methods trained on the real world ScalarFlow dataset. Lower values are better for all metrics under consideration.

| Algorithm | Input RMSE | Side RMSE | LPIPS | FID | Random FID | Inference time 120 steps |
|---|---|---|---|---|---|---|
| RapidGen (Qiu et al., 2021) | **.0227** | .0405 | .173 | 186 | 163 | 3m |
| Ours w/o $\tilde{\rho}$ | .0589 | .0670 | .191 | 170 | 123 | 4m |
| Ours | .0229 | **.0333** | **.116** | **134** | **102** | 4m |
| ScalarFlow (Eckert et al., 2019) | .0254 | .0488 | .152 | 153 | 119 | 6.5h |
| GlobTrans (Franz et al., 2021) | .0124 | .0281 | .085 | 102 | 77 | 37.5h |

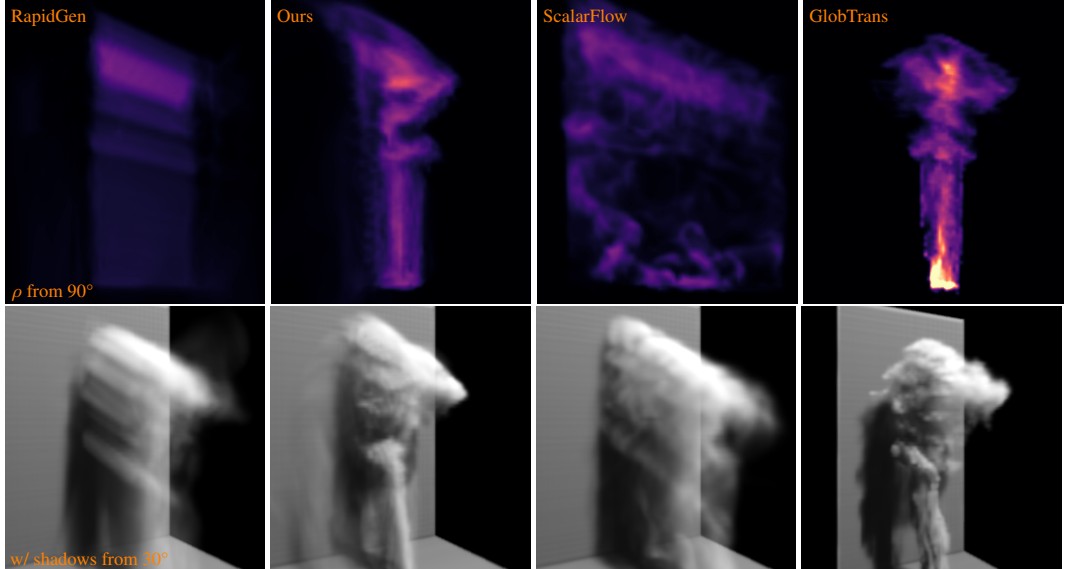

Figure 6: Visualization of density distributions of different approaches using ScalarFlow data for time-step 100. Our method shows similar structures as GlobTrans while being slightly less detailed. The single-scale RapidGan network fails to replicate the intricate 3D structures of real world data.

### 4.3.2 REAL-WORLD CAPTURES

The real-world data of the ScalarFlow dataset introduces the challenges of fine structures, sensor noise, and real-world backgrounds. It also does not provide measured 3D ground truth for comparisons, but the dataset still provides 5 calibrated views. With the center one being used as input, we use the remaining 4 to quantitatively evaluate the estimations from unseen views. Again, the input reconstruction in table 3 is good for all methods under consideration.

For the unseen side targets, our method with an FID of 134 now clearly outperforms RapidGen with 186, as well as the expensive ScalarFlow optimization (153). Only the GlobTrans optimization manages to do better in terms of this metric. We again evaluate the FID between random views of the estimated sequences and random samples of the dataset which clearly detects the lack of features in the RapidGen result. It also shows that structure-wise our method surpasses ScalarFlow, which takes ca. 97 times longer to produce a single output. While the GlobTrans optimization fares even better, it does so at the expense of a more than 560 times longer runtime. These measurements show that our trained network successfully generalizes to new inputs, and yields motion estimates that are on-par in terms of realism with classic, single-case optimization methods.

## 5 CONCLUSION

We presented *Neural Global Transport* (NGT), a new method to estimate a realistic density and velocity sequence from new single view sequences with a single pass of neural network evaluations. Our method and implementation currently have several limitations: e.g., it currently only supports white smoke, while anisotropic scattering, self-illuminaton, or the use of other materials would require an extension of the rendering model. Also, our transport model relies on advection without obstructions. Here, our method could be extended to support obstacles, or multi-phase flows. For even better long-term stability, the method could be extended to pass back information about future time steps for improved guidance.

Nonetheless, we have demonstrated that our networks can be trained with only single view data and short time horizons while still being stable for long sequences during inference. We solved the depth ambiguity arising from the single-view setting by using an adversarial loss, and we stabilized the velocity estimation by providing estimated prototype volumes both as input and as soft constraint. The resulting global trajectories of NGT are qualitatively and quantitatively competitive with single-scene optimization methods which require orders of magnitude longer runtimes.

## 6 REPRODUCIBILITY STATEMENT

Our source code is publicly available at `https://github.com/tum-pbs/Neural-Global-Transport` and includes the data and configurations necessary to reproduce all results of the paper. Further details about network architectures, training procedures and hyperparameters can also be found in the appendix.

ACKNOWLEDGMENTS

This work was supported by the Siemens/IAS Fellowship Digital Twin, and the DFG Research Unit TH 2034/2-1.

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

# APPENDIX

## A   TECHNICAL DETAILS

### A.1   NETWORKS

The full density generator $\mathcal{G}_\rho$ is detailed in figure 7, the velocity generator part $\mathcal{G}_{\mathbf{u}l}$ consists only of the 6 layers which are shared between all scales (see also figure 1). Like $\mathcal{G}_\rho$, it uses a kernel size of 5 and relu activation, except for the input and output convolutions which have a kernel size of 1. Any ResBlocks that change the number of filters from their input have a kernel size 1 convolution on their skip path. If the number of filters stays the same this convolution is omitted. We use the simple Layer Normalization of Xu et al. (2019) after every convolution except the last in a network.

**Discriminator**   We reuse the discriminator of Franz et al. (2021) which is a simple stack of convolutional layers. However, we remove the crop and rotation in the data augmentation to allow the discriminator to see the whole domain and consider the spatial orientation of observed features.

### A.2   LOSSES

**CFL magnitude loss**   For increased stability during advection we enforce the CFL condition on the velocity by using a magnitude loss that only affects velocity components larger than 1:

$$\mathcal{L}_{\text{CFL}} = \sum_{i=1}^{3} \max\left(\mathbf{u}_i^2 - 1, 0\right), \tag{11}$$

where $i$ are the x-, y-, and z-components of $\mathbf{u}$.

**Smoothness loss**   The first order smoothness loss is also defined per component:

$$\mathcal{L}_{\text{smooth}} = \sum_{i=1}^{3}\left[\left(\frac{\partial \mathbf{u}_i}{\partial x}\right)^2 + \left(\frac{\partial \mathbf{u}_i}{\partial y}\right)^2 + \left(\frac{\partial \mathbf{u}_i}{\partial z}\right)^2\right], \tag{12}$$

where the spatial derivative is approximated with finite differencing in practice.

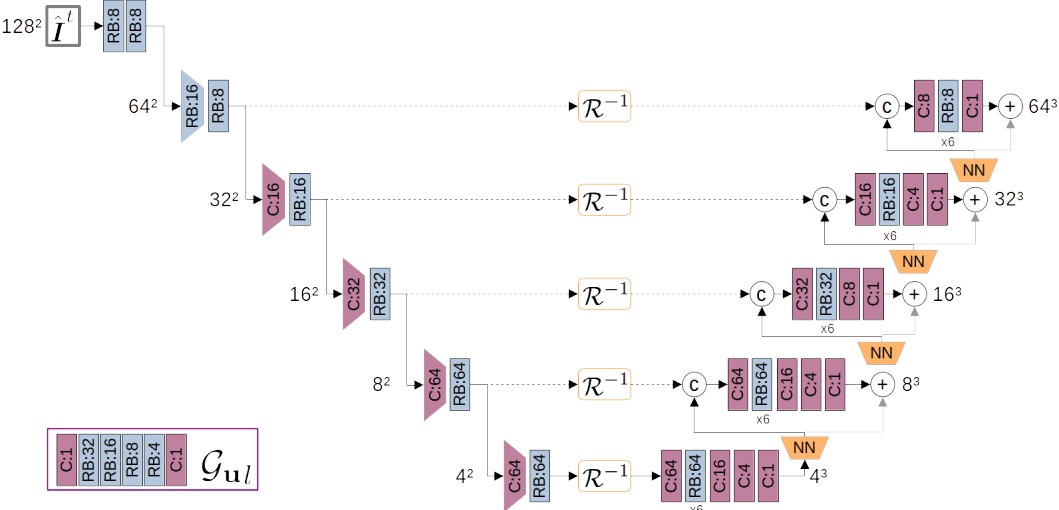

Figure 7: Our full density estimator network $\mathcal{G}_\rho$. Lower left: the velocity generator part $\mathcal{G}_{\mathbf{u}l}$ that is reused for each scale. C is a convolution, RB a ResBlock, NN a nearest neighbour sampling. A trapezoid shape indicate a stride of 2. Strided convolutions use a filter size of 4, the rest 5, all use ReLU activation. The number of filters and spatial resolutions are given in the image.

## A.3 TRAINING

Our method is implemented in TensorFlow version 1.12 under python version 3.6 and trained on a Nvidia GeForce GTX 1080 Ti 11GB. For training we use the Adam optimizer. For leaning rate decay we use the following formula:

$$\frac{\texttt{learning\_rate}}{1 + (\texttt{iteration} - \texttt{offset})\texttt{decay}} \tag{13}$$

**Data** Our synthetic dataset consists of 30 randomized simulations of buoyant smoke plumes, simulated with Mantaflow at a resolution of 128x192x128 (2x higher than our training resolution) for 150 time steps. We omit the first 45 frames as startup and train with the remaining 105 frames. The volumes are rendered with the same renderer we use during training to obtain the input images.

For the real world data we use the raw captures of first 30 scenes of the SalarFlow dataset as input images and omit the first 20 frames to ensure that there is enough density visible, leaving 130 frames per scene. We downsample the images by a factor of 10 to 192x108 to approximately match the projected grid resolution.

Each Network and method has been trained specifically for the corresponding dataset.

**Density training** $\mathcal{G}_\rho$ is trained with $\mathcal{L}_\rho = \mathcal{L}_{\hat{I}} + 2\text{e-}4\mathcal{L}_\mathcal{D} + 1\text{e-}3\mathcal{L}_z$ and a learning rate of 2e-4 with a decay of 2e-4, the decay is offset by -5000 iterations. We start at a grid resolution of 8x12x8 with 2 UNet levels. The resolution grows after 8k, 16k and 24k iterations by a factor of 2, adding a level of the UNet every time, thus reaching a maximum grid resolution of 64x96x64 with 5 levels. New levels are faded in over 3k iterations, starting 2k iterations after growth, by linearly interpolating between the up-sampled previous level and the current level. After fade-in only the output of the highest active level remains. The image resolution grows in conjunction with the density.

**Velocity training** For $\mathcal{G}_\mathbf{u}$ we use $\mathcal{L}_\rho = \mathcal{L}_{\hat{I}} + 1\text{e-}3\mathcal{L}_{\tilde{\rho}} + 2\text{e-}6\mathcal{L}_\mathcal{D} + 1\text{e-}3\mathcal{L}_z + 0.1\mathcal{L}_{\text{CFL}} + 1\text{e-}4\mathcal{L}_{\text{smooth}}$ with a learning rate of 4e-4 with the same decay of 2e-4, offset by -5000 iterations. The velocity training also starts at a grid resolution of 8x12x8 with 2 levels of $\mathcal{G}_{\mathbf{u}l}$ and with only 1 advection step (i.e. 2 frames). The density estimation and advection always happen at the maximum resolution of 64x96x64 and the velocity is up-scaled accordingly before it has reached this resolution. Before increasing the resolution, we extend the sequence, first to 3 frames after 1500 iterations and then to the final 5 frames after 3k iterations. Then we grow the resolution of the velocity estimation after 4k, 8k and 12k iterations, each time adding a new level of $\mathcal{G}_{\mathbf{u}l}$. The residual of a newly added velocity level is linearly faded in over 1500 iterations, starting 500 iterations after growth. A full training run of our model takes 6 days on average using a single GTX 1080 Ti GPU.

## A.4 RENDERING

We use the differentiable volumetric ray-marcher of Franz et al. (2021) for our method which discretizes the rendering equation

$$\mathcal{R}(\rho, L, c) = \int_n^f L(x) e^{-\int_n^x \rho(a)da} dx, \tag{14}$$

where $L$ gives the lighting at point $x$ and $c$ are camera parameters resulting in pixel rays from $n$ to $f$, by stepping along the ray to solve the integrals. The single-scattered lighting is realized by solving the rendering equation in the same way and storing intermediate step values to create the lighting volume $L$. A background image $\hat{I}_B$ can be added by attenuating it with the total density along the ray, thus the rendering becomes

$$\mathcal{R}(\rho, L, c) = \int_n^f L(x) e^{-\int_n^x \rho(a)da} dx + e^{-\int_n^f \rho(a)da} \hat{I}_B. \tag{15}$$

We modify the renderer to only render inside of the cell center of the outer cell-layer instead of their outer border to avoid border artifacts at the inflow, which are likely caused by the interpolation to 0 outside the domain.

## A.5 Projection into the volume

The operation $\mathcal{R}^{-1}$ transfers an image or 2D tensor into the volume by evenly distributing it along the pixel rays into the volume from a given view. This is achieved by inverting the ray-marching procedure. While marching along the ray the current pixel value is distributed to the 8 voxels around the current position, weighted by the distance to the voxel. The weights are the same one would use for linear interpolation at that point. Afterwards, the accumulated voxel-values are normalized with the number of contributions per voxel to obtain a smooth result. This procedure is also used for the gradient backwards pass of $\mathcal{R}$.

## A.6 Advection

The MacCormack advection (Selle et al., 2008) is a second order transport scheme using Semi-Langrangian advection as its base, which itself is implemented as a backwards-lookup using the velocity vectors and interpolating the transported field at the lookup location. With $\mathcal{A}_{SL}$ as the Semi-Langrangian base, MacCormac is defined as

$$
\begin{aligned}
\hat{\rho}^{t+1} &:= \mathcal{A}_{SL}(\rho^t, \mathbf{u}^t) \\
\hat{\rho}^t &:= \mathcal{A}_{SL}(\hat{\rho}^{t+1}, -\mathbf{u}^t) \\
\rho^{t+1} &:= \hat{\rho}^{t+1} + 0.5(\rho^t - \hat{\rho}^t).
\end{aligned}
\tag{16}
$$

To avoid extreme values the results are limited to be within the range of the interpolants encountered in the first Semi-Langrangian advection.

**Boundary and inflow handling**   All boundaries are open, meaning that advections that transport density from outside the domain use values from the nearest boundary layer. Inflow of densities is thereby automatically handled by the open boundaries which works well in our application but is theoretically limited as an initial density at the boundary layer is needed to have any inflow and the inflow can never exceed the already existing density values due to the linear interpolation in the advection.

**Vector potentials and curl**   Generating vector potentials and defining the velocity as their curl, as we do in our velocity estimation, can lead to discontinuous velocity fields if linear interpolation is used to interpolate vector potential fields between resolutions (Chang et al., 2021). Since the curl is based on spatial derivatives, and the derivatives of linear interpolation are not continuous, the resulting velocity can have discontinuities which can lead to grid aligned artifacts visible in the results. To circumvent this issue we use the higher order quadratic B-splices as interpolation kernels to transfer vector potentials from coarse to finer resolutions of our multi-scale network. This leads to a noticeably smoother velocity, as can be seen from the advected densities in figure 8.

# B Addtional results

## B.1 Ablations

We run additional ablations for the losses $\mathcal{L}_{\text{smooth}}$ and $\mathcal{L}_{\text{CFL}}$ when training $\mathcal{G}_{\mathbf{u}}$. The corresponding metrics are in table 4.

## B.2 Density prototypes

We visualize some of the density prototypes $\tilde{\rho}$ for the ScalarFlow dataset in figure 9. These are intended as a rough volumetric guidance via $\mathcal{L}_{\tilde{\rho}}$ and not seen as the real solution.

## B.3 ScalarFlow data

Additional results of our method evaluated on ScalarFlow data can be found in figures 10, 11 and 12 as well as in the supplemental videos.

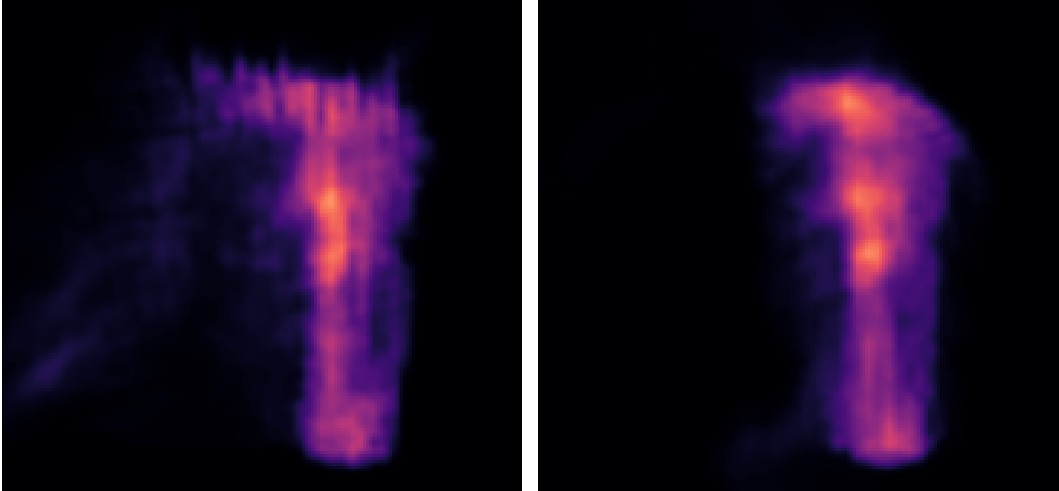

Figure 8: Left: the grid-aligned artifacts are caused by discontinuous velocities. Right: higher order interpolation of the potentials forces the velocities to be smoother, resulting in smoother densities after advection.

Table 4: Quantitative ablation of different regularization terms when training $\mathcal{G}_\mathbf{u}$ on the real world ScalarFlow dataset. The training diverges without any regularization (first row), leading to empty volumes. Without magnitude regularization ($\mathcal{L}_{\mathrm{CFL}}$, second row) the training itself is stable, but the evaluation produces velocities that quickly remove all density from the volume after a few time steps, again leaving empty volumes. Lower values are better for all metrics under consideration.

| Algorithm | $\mathcal{L}_{\mathrm{smooth}}$ | $\mathcal{L}_{\mathrm{CFL}}$ | $\mathcal{L}_{\tilde{\rho}}$ | Input RMSE | Side RMSE | LPIPS | FID | Random FID |
|---|---|---|---|---|---|---|---|---|
| Ours | | | | - | - | - | - | - |
| Ours | ✓ | | | - | - | - | - | - |
| Ours | | ✓ | | **.0227** | .0341 | **.117** | **136** | **84** |
| Ours | ✓ | ✓ | | .0589 | .0670 | .191 | 170 | 123 |
| Ours | ✓ | ✓ | ✓ | .0229 | **.0333** | **.116** | **134** | 102 |

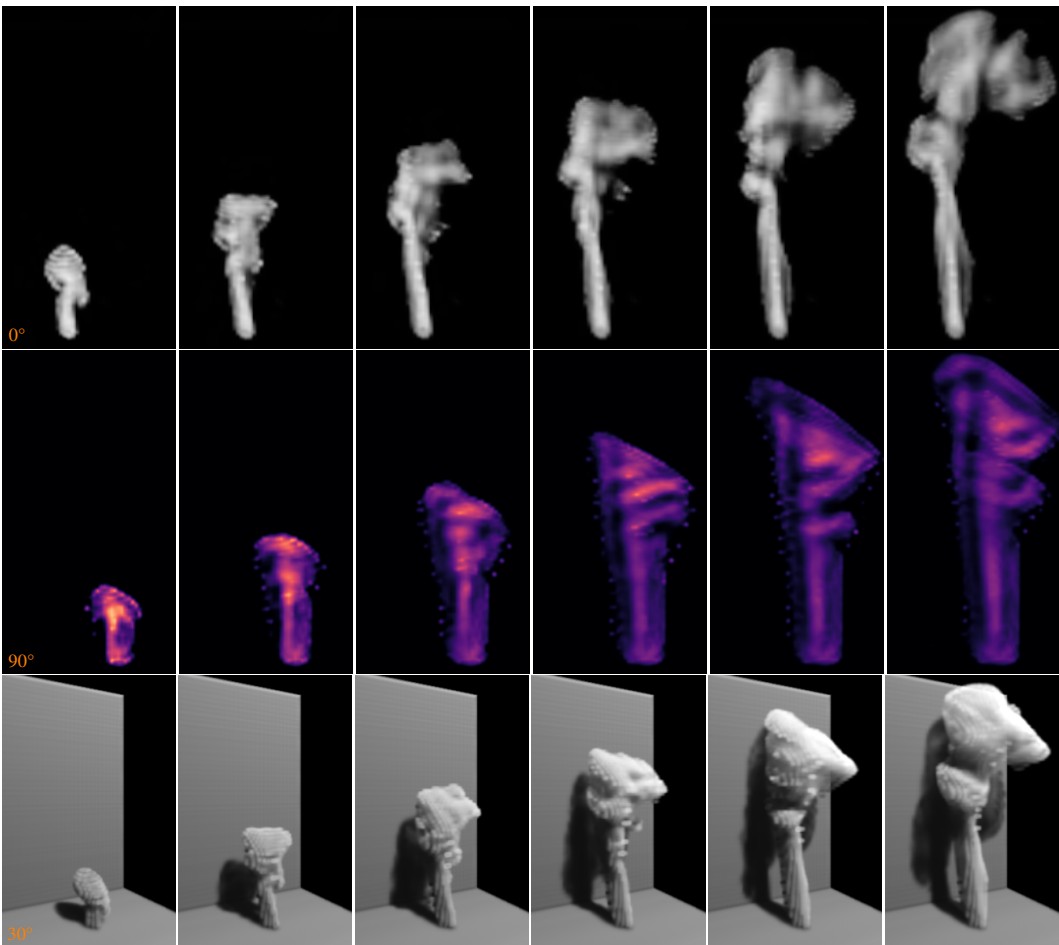

Figure 9: The density prototypes $\tilde{\rho}$ of ScalarFlow scene 80 used in the volumetric guidance $\mathcal{L}_{\tilde{\rho}}$ when training $\mathcal{G}_{\mathbf{u}}$. These are less detailed then the densities resulting from advection and show some artifacts around the border of the plume.

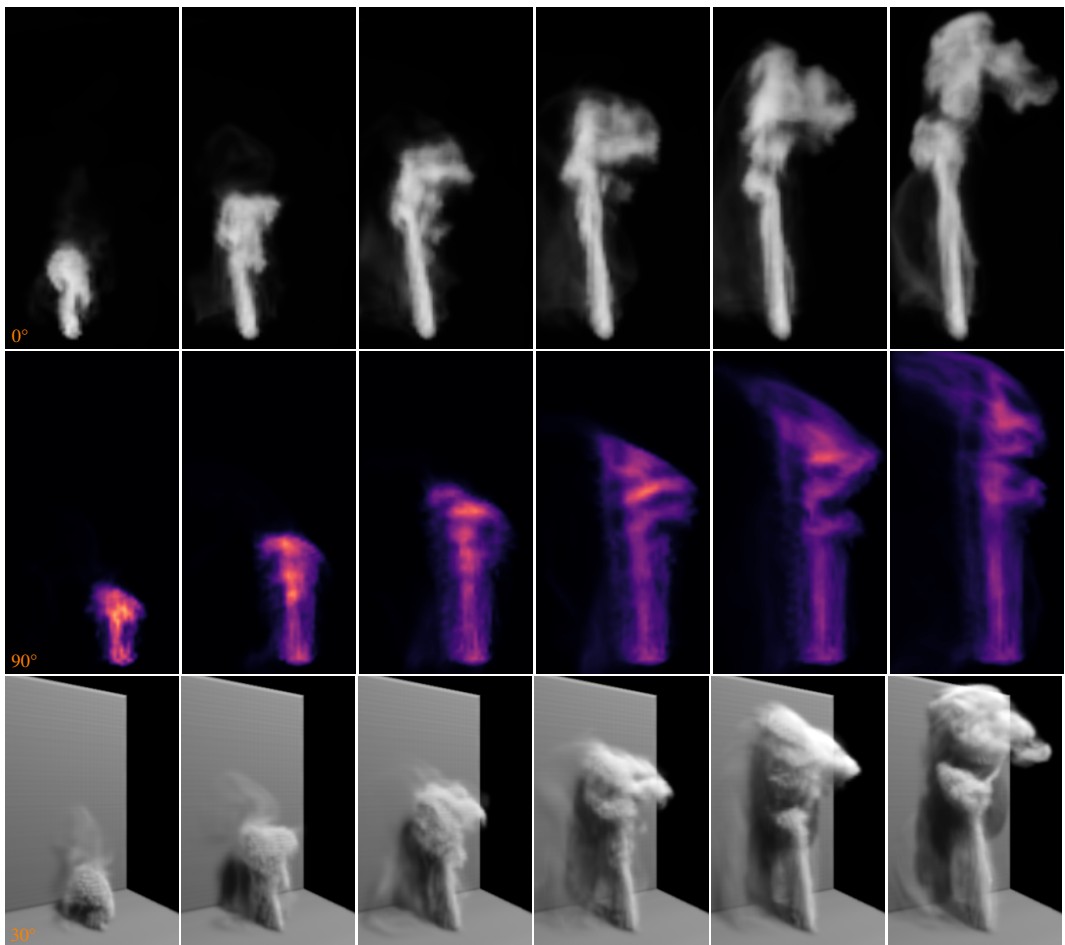

Figure 10: Our final model evaluated on ScalarFlow scene 80.

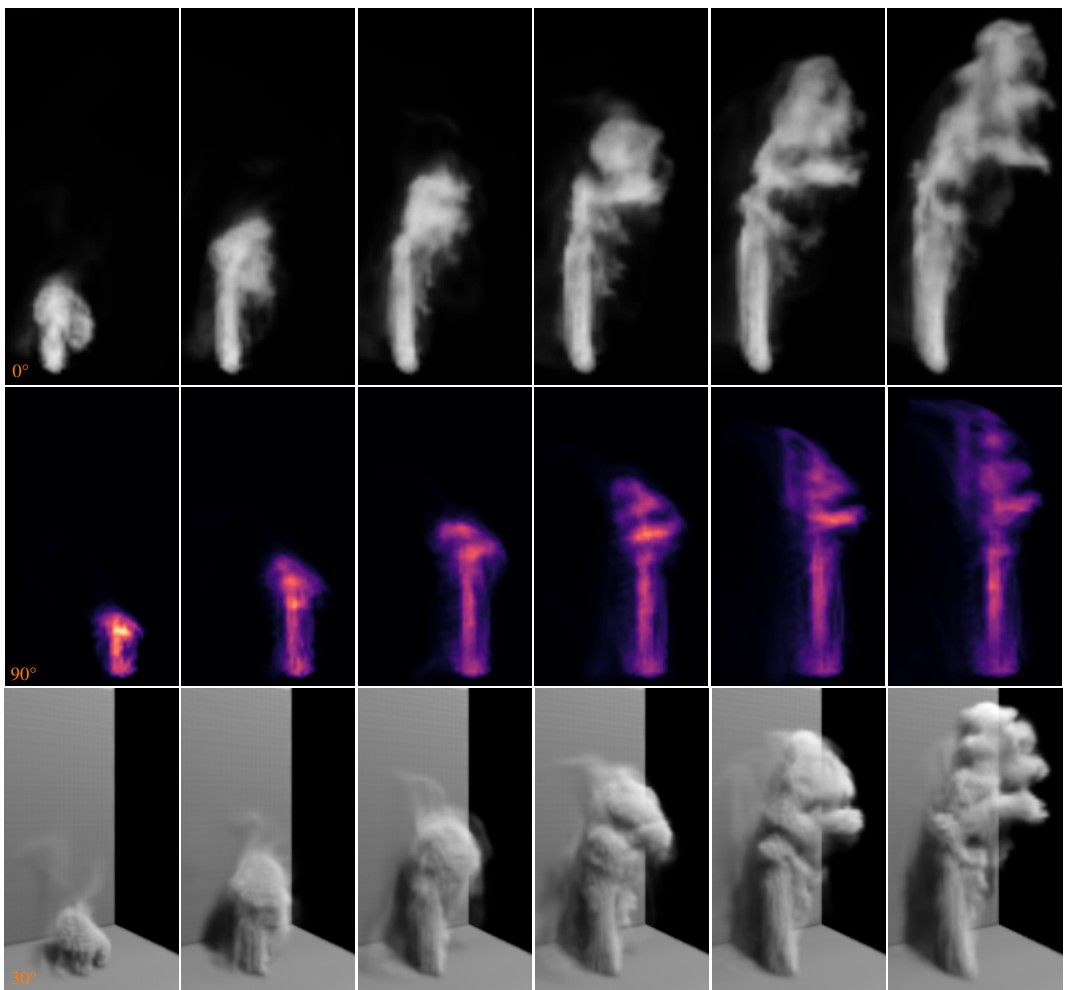

Figure 11: Our final model evaluated on ScalarFlow scene 81.

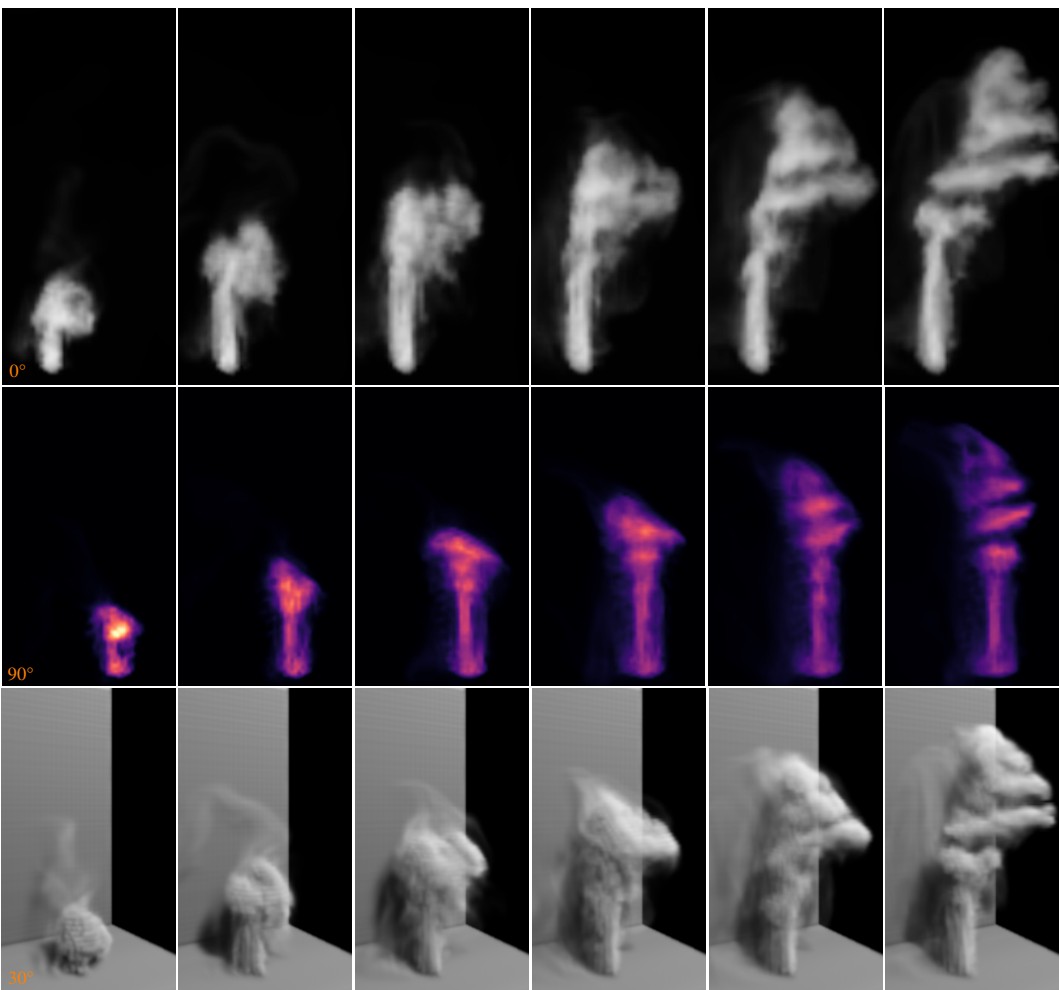

Figure 12: Our final model evaluated on ScalarFlow scene 82.

