# OpenReview forum: "Learning to Estimate Single-View Volumetric Flow Motions without 3D Supervision"
_ICLR.cc/2023/Conference — ICLR 2023 poster_

### Official Review · Reviewer_TTZh · 2022-10-24

**Confidence:** 4
**Correctness:** 3
**Technical Novelty And Significance:** 3
**Empirical Novelty And Significance:** 3
**Recommendation:** 8

**Clarity, Quality, Novelty And Reproducibility:**

### Clarity
As mentioned in the weakness part, the diagrams are hard to understand, and some technical designs lack background and intuition. The paper can be understood only by audiences familiar with all related work and techniques.

### Novelty
The contributions are significant and somewhat new. Aspects of the contributions exist in prior work.

### Reproducibility
The paper provides some implementation details. However, the training flow seems complicated as it requires multiple stages. To reproduce the experiment, more training details including the range of the hyperparameters tuned, and the final selected hyperparameter values are required.

**Strength And Weaknesses:**

### Strength
* The paper provides a reasonable explanation for the depth ambiguity problem in single-view inputs.
* Motion estimation for fluids is a challenging task, especially from single views. In general, the proposed method is technically sound to me.
### Weakness
* The framework figures are complicated and hard to understand. Too many notations, and some of them are unexplained. It would be better to simplify the diagrams and add some legends.
*  It would be better to give more background and intuitive explanations for some technical designs. For example, why use RaLSGAN objective to train the discriminator, and why would CFL condition help constrain the magnitude?
* It would be better to visualize the learned prototype volume.
* The ablation study is confusing. In Figure 3, there are two "fix pos, multi, ¬D"; one of them should probably be "var pos, multi, ¬D." It is unclear from Figure 3 that adding the adversarial loss can sharpen the result. It would be better to provide quantitative results for all ablation studies.
* The paper mentioned that using a multi-view capture setup does not help the density ambiguity issue and would result in a suboptimally averaged solution. How about using a multi-view+the proposed discriminator? Showing such a comparison could strengthen the evaluation.
* Some minor typos. Section 3.1 - "leadsto",  Section 4.1 - "spacial".

**Summary Of The Paper:**

The paper proposes a volumetric motion estimation framework specifically in the context of fluids. The framework can be trained end-to-end from only single-view image sequences. Experiments show that the proposed framework can successfully generalize across different inputs and outperforms single-scene optimization-based approaches.

**Summary Of The Review:**

In general, the proposed method is technically sound to me. However, the experiments are somewhat limited. Furthermore, the clarity of the paper needs to be improved. Please see the above sections for more details.

---

> ### Author Response · Authors · 2022-11-18
> **Response to Reviewer TTZh**
>
> Thank you for your constructive feedback.
>
> **Framework figures**
> We agree that the previous version was unnecessarily complicated. We have simplified Fig. 1 as much as possible without omitting any essential parts, and added a legend for the colors used. The removed details are explained in the appendix now. Fig. 2 was removed in favor of the complete $G_\rho$ diagram in the appendix, which also contains explanations for the symbols.
>
> **Background and intuitive explanations**
> We use a simple magnitude loss on velocities above 1 to implement a CFL condition.
> This is done for increased stability in the advection. We clarified that section of the paper and added the exact loss formulas to the appendix.
> GAN balancing and stability is usually very difficult, so we opted to use the established RaLSGAN which was successfully used in previous work for a similar task.
>
> **Learned prototype volumes**
> It is a good idea to show the prototypes. We added a visualization of the density prototype to the appendix and the supplemental videos to give some intuition for the type of volumes that are inferred.
>
> **Ablation study**
> We added a quantitative evaluation of the ablation in table 1 and extended the discussion in section 4.1. It clearly shows that the discriminator perceptually improves the results, especially when no additional targets are present. E.g., the FID improves from 98 to 68 here.
> %That the discriminator is unable to find the correct location of the object is evident from the still high density error of 0.93.
> The density error of 0.93 simply indicates that the discriminator by itself has to way of determining the ground truth depth location.
>
> **Multi-view image loss in combination with Discriminator**
> We have added an additional ablation for the case of multi-view targets + discriminator to fig. 2 and table 1 and integrated it into the discussion in section 4.1.
> Due to the problem of depth ambiguity, the additional targets add constraints that appear random given only the input view, as $G_\rho$ can not infer the specific target used from the available input.
> Even when combined with the discriminator, the multi-view targets push the generator towards averaged solutions which conflicts with the GAN objective of finding a specific solution.
> In the metrics this becomes evident:
> using multiple targets leads to a lower density RMSE, as RMSE typically favours averaged solutions.
> However, the perceptual errors improve (LPIPS from 0.124 to 0.114 and FID from 80 to 68) when removing the additional targets in the presence of the discriminator.
> We'd also like to point out that using multi-view targets is very challenging in real world settings as it requires a calibrated capture setup. Hence, we believe the single-view approach is very attractive despite the ambiguity of the solution manifolds.

---

> > ### Comment · Reviewer_TTZh · 2022-12-09
> > **Response to Authors**
> >
> > Thank you for the response. My concerns are all well-addressed in the revision; therefore, I would recommend acceptance.

---

### Official Review · Reviewer_8YBe · 2022-10-24

**Confidence:** 4
**Correctness:** 3
**Technical Novelty And Significance:** 2
**Empirical Novelty And Significance:** Not applicable
**Recommendation:** 6

**Clarity, Quality, Novelty And Reproducibility:**

Quality and clarity:
The clarity needs to improve (see the weakness above for some confusions)
Additional ablation studies on the regularizations are needed.

Originality :
Although the monocular 3D and scene flow estimation is not a new topic, the inclusion of the MacCormack Advection to model the inter-frame smoke transport is incrementally novel

**Strength And Weaknesses:**

Strength

+The usage of the transport function A() based on MacCormack advection  to render new frames adds physical inductive bias into the pipeline, so that the estimated states (density) can be propagated into the new timestamp in a physically correct way. It also helps to reduce the number of learnable parameters for the transition itself (eg. versus using GRU) and make the model more generalizable.

+ The adversarial learning helps to reduce the estimation ambiguity.

+ The pipeline trained end-to-end without GT volumetric density and velocity as supervisions, which is useful since GT volumetric data is hard to capture in real world.

Weakness

-Key assumption not explained: The rendering equation does not include the background color. If the background color changes, how to make sure this rendering equation is correct? If there is an additional assumption about the colors of the background and smoke, it should be claimed upfront since it is a very strong assumption

-The writing needs improvement:
What does it mean by c_z is the center along z (after Eq.6)? Without defining the start and end points along z for each ray, it is hard to understand it.
Eq.6: what is p? Is it a 3D point location? If so, how can we define L_z over one single 3D point? *
Eq.6: why multiplying 2 inside the second term eq.6 what is the intuition of the regularization?
What is R^{-1} function in Eq.8

-To tackle the ambiguities in the solution, the method assumes  the initial state is known. What if the initial state is not accurate? How can the subsequent estimation correct for the inaccurate initial state estimation?

-Ablation study is not enough: ablations for the additional regularizations that tackle ambiguities are missing.


**Summary Of The Paper:**

This paper proposes a single-view video-based volumetric reconstruction pipeline for smoke. The method assumes known background (black color) and foreground (smoke with known color). It takes the video frames as input and estimates both volumetric density and velocity. The pipeline is trained end-to-end by comparing the captured incoming frame with the rendered image that is based on the estimated volumetric density and velocity. Additional regularizations are added to tackle the inherent ambiguity in single view volumetric estimation.

**Summary Of The Review:**

The paper demonstrates a monocular video based method for smoke volumetric density and velocity reconstruction. Although the method has shown to be working for smoke with strict experiment setup (known foreground and background colors), I’m not sure if it can be extended to handle more general case (eg. smoke with real-world background, or even non smoke foreground). In addition, due to the confusions in the writing, the motivation for the design choice is not clear to me. Last but not least, the experiment is lacking in ablations. As a result, I would recommend weak rejection of the submission.

---

> ### Author Response · Authors · 2022-11-18
> **Response to Reviewer 8YBe**
>
> We thank the reviewer for the detailed and helpful comments.
>
> **Rendering and background**
> Thank you for pointing this out.
> While the background is assumed to be known, it is provided in terms of a simple image. This background handling was always part of the rendering and image loses, and is now clearly explained in our revised version.
> We mention the use of background images now in the corresponding sections of the paper and added the details about the background blending to the appendix.
>
> **Meaning of Eq. 6**
> The intuition behind Eq. 6 is to keep the estimated density in the center of the volume domain to obtain a more compact result, i.e. we refer to these terms as the center loss.
> This is done by increasingly penalizing density the further its location is from a designated center $c_z$.
> In general, $c_z$ is a problem specific distance from the camera, for which we choose the center point of the reconstruction domain.
> $L_z$ is defined for every point $p$ of the volume, penalizing the density at that point ($\rho(p)$), weighted with its distance to the center. This distance is computed along the primary camera view direction, between $c_z$ and the projected point $p_z$. The distance is further normalized with half the grid resolution $r$, s.t. it is 1 at the domain boundaries.
>
> **Projection operator $R$ and its inverse**
> $R^{-1}$ essentially distributes an image (or any 2D feature tensor) into the volumes along the pixel rays of a given camera or view.
> This is done by "inverse" ray-marching: marching along the ray, the pixel-value is added to the surrounding voxels at each step (which would be used for interpolation in the forward rendering). These scattered values are then normalized by the distance from the ray (using what would be the interpolation weights in the forward rendering) and the number of contributions per voxel to obtain a smooth result.
> We clarified that before Eq. 8 and added an explanation in the appendix.
>
> **Initial state accuracy**
> The initial state $\rho^0$ is not known or provided, but estimated by the network $G_{\rho}$. We assume it to be known in the beginning of the Method-section for the sake of the argumentation, and section 3.3 explains that it is now inferred by $G_{\rho}$.
> Currently, the initial state can not be changed once estimated, but the velocity estimation typically has enough degrees of freedom to correct for the inaccuracies via transport and inflow at the boundaries. We found that the exact shape of the initial density guess is not overly important as the velocity can deform the density at each step and thus compensate for errors.
>
> **Additional ablations**
> New ablations for the density regularization can now be found in Fig. 2, for which we added a quantitative evaluation to the paper in Tab. 1.
> An ablation for the volume prototype can be found in Fig. 3, the corresponding metrics are in Tab. 3.
> We will additionally include ablations for the remaining regularization terms, namely magnitude (CFL) and smoothness regularization for the velocity, in a future version of our submission.

---

> > ### Comment · Reviewer_8YBe · 2022-12-09
> > **reply to the rebuttal**
> >
> > The authors made substantial revision of the submission, adding experiments on ablations and making the method description more clear. My concerns regarding the initial state inaccuracy, background image handling and missing ablations are addressed in the revision. Now the submission is clear and convincing enough.  As a result, I'm willing to raise my score to weak acceptance.

---

### Official Review · Reviewer_UynW · 2022-10-27

**Confidence:** 3
**Correctness:** 3
**Technical Novelty And Significance:** 3
**Empirical Novelty And Significance:** 3
**Recommendation:** 6

**Clarity, Quality, Novelty And Reproducibility:**

The writing and clarity of the paper can be improved.

- Notation: this paper uses a large amount of notation. Some are not properly defined. For example, in Figure 1, the loss L_u is not defined in the paper. Also, the notation \nabla\times is not explained.
- Consistency: For example, in Eq.(3), the input to network G_u includes \rho^t, I^{t+1}, but it is later shown in Eq.(8) that the input to G_u consists of five terms. This is quite confusing.
- Grammar errors and typos:
page 1: "is a very appealing"
page 3: "leadsto"
- I also suggest that some long sentences to be broken into shorter sentences to improve the readability.

Reproducibility: the authors indicates that code and data will be made public upon paper acceptance.

**Strength And Weaknesses:**

Strength:

+ A novelty of this paper is that it trains deep networks to predict 3D flow and volumetric densities in an end-to-end fashion. Compared to optimization-based methods, the proposed method is much faster.
+ Technically, the proposed method appears to be sound and effective. It adopts an adversarial loss, similar to the one used in [Franz et al., 2021], to regularize the estimated densities across different views. To some extent, this helps avoid requiring videos from multiple views as input, as prior work [Qiu et al., 2021] does. It also proposes to predict a prototype volume, which is shown to be beneficial in stabilizing the network training over time.

Weaknesses:

- The quality of the estimated flow is not very good. In Table 1, the errors in estimated flow u is much larger than the other methods on the synthetic dataset. Looking at the videos in the supp material, there are also noticeable artifacts in the flow motion. Perhaps, the design of network and losses makes the proposed method focus more on synthesizing the densities \rho than predicting the flow u?
- What are the limitations of this work? This is not discussed in the paper.

**Summary Of The Paper:**

This paper presents a new method to estimate the 3D flow and volumetric densities of a moving fluid from a monocular video. The proposed framework consists of two deep networks, one for predicting the volumetric densities from a single image, and one for predicting flow given the input images and predicted densities from two consecutive timestamps. Since ground truth 3D flow and densities are difficult to obtain, this paper adopts an unsupervised learning approach, which projects the inferred densities to the image space with a differentiable image formation operator, and then uses image-based losses to train the networks. To reduce the depth ambiguity, it further uses an adversarial loss to regularize the solution. Experiment results on benchmark datasets demonstrate the effectiveness and efficiency of the proposed method.

**Summary Of The Review:**

Overall, this work presents an effective and efficient solution to the challenging problem of estimating 3D flow and volumetric densities from a monocular video. Leveraging unsupervised learning and adversarial losses appears to be a promising direction. However, the quality of the estimated 3D flow is not as good. To improve the results, more work needs to be done to adjust the network and losses.

---

> ### Author Response · Authors · 2022-11-18
> **Response to Reviewer UynW**
>
> We thank the reviewer for the constructive comments and the positive assessment of our submission.
>
> **Volumetric velocity errors**
> It is worth noting that our work does not try to match a single, specific reference but rather addresses the estimation of an unobserved quantity in a chaotic and severely under-constrained setting.
> Thus, the high velocity error can be explained by the relatively unconstrained velocity and the multitude of solution that can lead to a sequence matching the input view but deviate from any specific reference velocity.
> Furthermore, velocities in regions without density can be arbitrary without changing the observed result.
> The estimation can also be shifted in the volume, for which the simple error metrics do not account, leading to an artificially high error.
> After all, in our setting of single-view estimation, the unobserved velocity has no direct constraints (other than some regularization) and any velocity that leads to a density matching the image-based constraints is acceptable. We have clarified this in section 4.3.1.
> The FID and matching input images in terms of RMSE are thus more suitable for evaluation here. The comparison to GT velocities was added for completeness, but is definitely problematic in our setting.
> Rather, the high velocity error, in combination with the lower volume and image errors, are a further indication that multiple velocity solutions that lead to similar density observations exist.
> As for the artifacts, we already achieved noticeable improvements by using a higher order interpolation kernel when upsampling velocity potentials, but we believe that there is potential for further hyperparameter tuning to improve the synthetic results.
>
> **Artifacts in the flow**
> We removed these artifacts by switching to a higher order interpolation kernel when up-scaling the velocity potentials. They were not related to the core of the proposed method. Please see the general comment above for the explanation behind this.
>
> **Limitations**
> We use a simplified rendering model that supports isotropic single scattering with white densities.
> While the use of background images is supported, obstructions, i.e. objects in front of the volume, or objects inside the flow are not (yet) supported.
> These limitations are now discussed in section 5 of our submission.
> Furthermore, the paper primarily focuses on rising smoke plumes, other types of flows or inputs from particle image velocitemtries would be an interesting direction for future work.
>
> **Notation**
> We removed the unused loss-notation from Fig. 1.
> $\nabla\times$ denotes the curl operator which we clarified in the revised paper.
>
> **Consistency**
> We extend the input to $G_u$ from the initial version with any unambiguous input available to help guide the velocity estimation. More explanations can be found in the paragraph above Eq. 8.

---

> > ### Comment · Reviewer_UynW · 2022-12-10
> > **Reply to the rebuttal**
> >
> > The authors have addressed my concerns regarding the artifacts and limitations in the revised paper and the rebuttal. I am happy to recommend acceptance.

---

### Author Response · Authors · 2022-11-18
**General Response**

We want to thank all reviewers for their efforts and constructive feedback.

We have uploaded a revised version with highlighted changes where we have updated our results and revised several sections for increased clarity. We also added more information and additional comparisons to the appendix.

Specifically, we have addressed the grid-aligned artifacts present in both our synthetic and ScalarFlow results. These artifacts are the result of discontinuities in the velocities at cell boundaries, caused by the combination of linear interpolation and curl. Since the curl is defined as differences of spatial derivatives, applying it to a piecewise linear field results in discontinuities.
The revised version of our paper includes results using quadratic B-splines for interpolation, which removes these artifacts.
No other changes where made to our method: only $G_u$ was trained anew with the B-spline interpolation, retaining all original hyperparameters. This update improves the metrics on the ScalarFlow data across the board, with the random view FID improving from 117 to 102.

We give more detailed answers to the individual questions of reviewers below.

---

### Decision · Program_Chairs · 2023-01-20

**Decision:**

Accept: poster

**Justification For Why Not Higher Score:**

Very specific application area.

**Justification For Why Not Lower Score:**

Still good results and novel formulation.

**Metareview: Summary, Strengths And Weaknesses:**

The paper demonstrates a monocular video based method for fluid and smoke volumetric density and velocity reconstruction.  The proposed framework consists of two deep networks, one for predicting the volumetric densities from a single image, and one for predicting flow given the input images and predicted densities from two consecutive timestamps. Since ground truth 3D flow and densities are difficult to obtain, this paper adopts a self-supervised learning approach, which projects the inferred densities to the image space with a differentiable image formation operator, and then uses image-based losses to train the networks. To reduce the depth ambiguity, it further uses an adversarial loss to regularize the solution. Experimental results on benchmark datasets demonstrate the effectiveness and efficiency of the proposed method.
The reviewers raised concerns regarding the quality of the estimated flow depicted in the supplementary video and
motivation for the rendering equations used, which the rebuttal supplied by the authors successfully addressed. The paper has been significantly strengthened post rebuttal.


**Note From Pc:**

if the above contains the word "oral" or "spotlight" please see: "oral" presentation means -> notable-top-5% and "spotlight" means -> notable-top-25%. As stated in our emails, we are disassociating presentation type from AC recommendations

**Summary Of Ac-Reviewer Meeting:**

N/A